# Intestinal parasitic infection among household contacts of primary cases, a comparative cross-sectional study

**Berhanu Elfu Feleke** [1] *, **Melkamu Bedimo Beyene**[1], **Teferi Elfu Feleke**[2], **Tadesse Hailu Jember**[3], **Bayeh Abera**[4]

**1** Department of Epidemiology and Biostatistics, University of Bahir Dar, Bahir Dar, Ethiopia, **2** Department of Pediatrics, St Paul University, Addis Ababa, Ethiopia, **3** Department of Medical Laboratory Science, College of Medicine and Health Sciences, Bahir Dar University, Bahir Dar City, Ethiopia, **4** Department of Microbiology, College of Medicine and Health Sciences, Bahir Dar University, Ethiopia

* elfufeleke@gmail.com

## Abstract

### Background

Intestinal parasitic infection affects 3.5 billion people in the world and mostly affecting the low socio-economic groups. The objectives of this research works were to estimate the prevalence and determinants of intestinal parasitic infection among family members of known intestinal parasite infected patients.

### Methods and materials

A comparative cross-sectional study design was implemented in the urban and rural settings of Mecha district. The data were collected from August 2017toMarch 2019 from intestinal parasite infected patient household members. Epi-info software was used to calculate the sample size, 4531 household members were estimated to be included. Data were collected using interview technique, and collecting stool samples from each household contact of intestinal parasite patients. Descriptive statistics were used to estimate the prevalence of intestinal parasites among known contacts of intestinal parasite patients/family members. Binary logistic regression was used to identify the determinant factors of intestinal parasitic infection among family members.

### Results

The prevalence of intestinal parasite among household contacts of intestinal parasite-infected family members was 86.14% [95% CI: 86.14% - 87.15%]. *Hookworm* infection was the predominant type of infection (18.8%). Intestinal parasitic infection was associated with sex, environmental sanitation, overcrowding, personal hygiene, residence, substandard house, role in the household, source of light for the house, trimmed fingernails, family size, regular handwashing practice. Protozoa infection was associated with habit of ingesting raw vegetable, playing with domestic animals, water source and the presence of household water filtering materials.

**Data Availability Statement:** All relevant data are within the manuscript and its Supporting Information files.

**Funding:** This research work was financially sponsored by Bahir Dar University and the funders had no role in study design, data collection and analysis, decision to publish, or preparation of the manuscript.

**Competing interests:** The authors have declared that no competing interests exist.

## Conclusion

High prevalence of intestinal parasitic infection was observed among household contacts of primary cases.

## Introduction

Intestinal parasites are groups of worm's primary affecting the gastrointestinal tracts, they broadly contain flatworms (tapeworms and flukes) and roundworms (Ascariasis, Pinworm, and Hookworm infections) [1]. The mode of transmission includes ingestion of uncooked animal products, consuming infected water, absorption through the skin and feco-oral routes [2] [3]. That means all family members living in intestinal parasite-infected patients are at higher risk of acquiring the infection.

Abdominal cramp, vomiting, excessive bowl sound, nausea, diarrhea, loss of appetite, malabsorption, skin itching are some of the manifestations of intestinal parasites [4].The diagnosis of intestinal parasitic infection usually performed by taking stool samples and applying different laboratory techniques, concentration technique is more valid than the other laboratory techniques [5].

Intestinal parasitic infection affects 3.5 billion people in the world and mostly affecting the low socio-economic groups [6]. Soil-transmitted helminths infection (Ascaris Lumbricoides, Trichuris trichiura, and Hookworm) alone affects 1.5 billion people worldwide [7]. Sub-Saharan Africa bears the highest-burden for both helminths infection and other intestinal parasitic infections [8].

Intestinal parasitic infection can be complicated with malnutrition, intestinal obstruction, growth retardation, immunodeficiency and affecting the socioeconomic development of the nations [9].

Gender, age, role in the household, socioeconomic characteristics, levels of education, poor sanitation, proximity to water sources, family size, environmental sanitation, handwashing practice, untrimmed fingernail, housing conditions, resident, barefoot are some of the predictors of intestinal parasitic infection [10–18].

The management of intestinal parasitic infection was not complicated and most intestinal parasitic infection can be effectively treated with a single dose of anti-helminths. However, the intestinal parasitic intervention neglects the household contacts because there is no available evidence on the prevalence of intestinal parasites among household members; so, this study was conducted to give baseline evidence on the estimate of household secondary cases.

The objectives of this research works were to estimate the prevalence and determinants of intestinal parasitic infection among family members of known intestinal parasitic infected patients.

## Methods and materials

The comparative cross-sectional study design was implemented in the urban and rural settings of mecha district. Mecha district is located in the north-west of Ethiopia. In the district; there are 10 health centers and 1 general hospital. The data were collected from August 2017 to March 2019. Data were collected from intestinal parasitic infected patient household members.

The sample size was calculated using Epi-info software version 7 using the assumption of 95% CI, power of 85, rural to an urban ratio of 2:1, the none response rate of 10% gives an estimated 1510 household members from the urban setting and 3021 household members from the rural settings.

Household members were selected using contact tracing. A patient diagnosed positive for the parasitic infection in the district health facility was used to trace for their family members intestinal parasitic infection status. Intestinal parasitic infection status was checked from all household contacts.

Interview technique and laboratory methods were used to collect the data. Clinical nurses were recruited for the interview and health officers were recruited for the supervision. The stool sample was collected from each family member of known intestinal parasitic infected patients and transported to the nearby health facility for the analysis. From each known contact, one gram stool sample was collected in 10 ml SAF (sodium acetate- acetic acid-formalin solution). Direct microscopic stool examination and concentration techniques were used. Formal ether concentration technique was used to identify the presence of intestinal parasites. The stool sample was well mixed and filtered using a funnel with gauze. Around 7 ML (Milliliter) normal saline and 3 ml of ether was added, mixed well and then centrifuged for 5 minutes at 2000 RPM. Finally, the supernatant was discarded and the sediment was examined for parasites under the microscope [19].

Data were entered to Epi-info software and transported to SPSS for the analysis. Descriptive statistics were used to estimate the prevalence of intestinal parasites. Binary logistic regression was used to identify the determinant factors of intestinal parasitic infection among family members. Handwashing practice was measured if the participants wash his/her hands after visiting the toilet, before cooking food and before feeding.

Ethical clearance was obtained from Bahir Dar university ethical review board of (ethical approval number የኢ.ህ.ጤቴ/ሽ/ዓ/01/795). Permission letter was obtained from Amhara National Regional State Health Bureau ethical committee and Mecha district health office. Written informed consent was obtained from each study participant or guardians. Those study participants with intestinal parasites were referred to the nearby health facility for further management. The confidentiality of the data was kept at all stages.

## Results

A total of 4436 study participants were included giving for the response rate of 98%, 64 study participants were unwilling to participate in the study and 31 study participants were excluded due to poor quality of stool sample. Female constitute 50% of the study participants, and 67% of the study participants were from the rural area. (Table 1)

The prevalence of intestinal parasitic infection among family members was 86.14% [95% CI: 86.14% - 87.15%]. *Hookworm* infection (18.8%) was the predominant parasitic infection followed by *Enatmeba histolytic/dispar* (11.4%), 36.2% of family members had heavy intensity of infection (Table 2).

### Intestinal parasitic infection among children

The prevalence of intestinal parasitic infection among children family members was 82.77% [95% CI: 81.08% -84.47%]. After adjusting for sex, environmental sanitation, source of light for the house, size of the fingernails, family size, overcrowding, personal hygiene, the presence of chicken in the house, and substandard house: Intestinal parasitic infection among household members was associated with sex, environmental sanitation, the presence of chicken in the house, overcrowding, personal hygiene, residence, and substandard house (Table 3)

### Intestinal parasitic infection in adult household members

The prevalence of intestinal parasitic infection among household members whose age greater than 16 years was 88.67% [95% CI: 87.43% -89.90%]. After adjusting for sex, role in the

**Table 1. Population profile of the study participants (n = 4436).**

| Serial number | Population profile | | Frequency | Percentage |
|---|---|---|---|---|
| 1. | Sex | Female | 2206 | 49.7 |
| | | Male | 2230 | 50.3 |
| 2. | Environmental sanitation | Clean | 1323 | 29.8 |
| | | Dirty | 3113 | 70.2 |
| 3. | Source of light for the house | Modern | 1073 | 24.2 |
| | | Traditional | 3363 | 75.8 |
| 4. | Floor materials of the house | Mud | 3190 | 71.8 |
| | | Others | 1246 | 28.2 |
| 5. | Household water filtering mechanisms | Present | 861 | 19.4 |
| | | Absent | 3575 | 80.6 |
| 6. | Fingernails of the respondents | Trimmed | 927 | 20.9 |
| | | Not trimmed | 3509 | 79.1 |
| 7. | Family size | ≤4 | 661 | 14.9 |
| | | >4 | 3775 | 85.1 |
| 8. | Educational status | Illiterate | 1744 | 39.3 |
| | | Formal education | 2557 | 57.6 |
| | | Informal education | 135 | 3 |
| 9. | Resident | Rural | 2960 | 66.7 |
| | | Urban | 1476 | 33.3 |
| 10. | Marital status | Single | 3320 | 74.8 |
| | | Married | 1056 | 23.8 |
| | | Divorced | 42 | 0.9 |
| | | Widowed | 18 | 0.4 |
| 11. | Age in years | 0–10 | 1744 | 39.3 |
| | | 11–20 | 2035 | 45.9 |
| | | 21–30 | 215 | 4.8 |
| | | 31–40 | 303 | 6.8 |
| | | 41–50 | 12 | 0.3 |
| | | >50 | 127 | 2.9 |

household, environmental sanitation, source of light, trimmed fingernails, substandard house, family size, the presence of chicken in the house, handwashing behavior, overcrowding, personal hygiene, residence and chronic illness: intestinal parasitic infection among household

**Table 2. The type of parasitic infection among household members (n = 4436).**

| Intestinal parasitic species | Frequency | Percent |
|---|---|---|
| Not infected | 615 | 13.9 |
| *Hookworm* | 834 | 18.8 |
| *Ascaris lumbricoides* | 375 | 8.5 |
| *S. mansoni* | 198 | 4.5 |
| *Trichuris trichiura* | 332 | 7.5 |
| *E. histolytica/dispar* | 505 | 11.4 |
| *Balantidium coli* | 411 | 9.3 |
| *G. lamblia* | 302 | 6.8 |
| *Hymenolepis nana* | 29 | .7 |
| Mixed infections | 835 | 18.8 |

**Table 3. The determinants of intestinal parasitic infection among children household members (n = 1904).**

| Variable | | IP | | COR [95% CI] | AOR [95% CI] | p-value |
|---|---|---|---|---|---|---|
| | | Infected | Not infected | | | |
| **Sex** | Male | 717 | 168 | 0.79 [0.62–1.02] | 0.76[0.58–0.99] | 0.04 |
| | Female | 859 | 160 | | | |
| **Environmental sanitation** | Clean | 168 | 10 | 3.79 [1.92–7.71] | 0.04 [0.01–0.14] | <0.01 |
| | Dirty | 1408 | 318 | | | |
| **Chicken in the household** | Present | 1069 | 256 | 0.59 [0.44–0.79] | 4.42 [2.81–6.95] | <0.01 |
| | Absent | 507 | 72 | | | |
| **Overcrowding** | Present | 956 | 152 | 1.79 [1.40–2.28] | 2.14 [1.6–2.88] | 0.01 |
| | Absent | 620 | 176 | | | |
| **Personal hygiene** | Clean | 1395 | 312 | 0.4 [0.22–0.68] | 0.26 [0.07–0.93] | 0.04 |
| | Not clean | 181 | 16 | | | |
| **Resident** | Urban | 576 | 92 | 1.48 [1.13–1.94] | 2.68 [1.86–3.89] | <0.01 |
| | Rural | 1000 | 236 | | | |
| **Substandard house** | Yes | 237 | 42 | 1.21 [0.84–1.74] | 1.92 [1.03–3.6] | 0.04 |
| | no | 1339 | 286 | | | |

members were associated with sex, role in the household, environmental sanitation, source of light, trimmed fingernails, substandard house, family size, the presence of chicken in the house, regular handwashing practice, personal hygiene, and resident (Table 4).

The odds of soil transmitted helminths among barefooted individuals were 1.51 folds higher. Habit of ingesting raw vegetables increases the odds of protozoa infection by 2.96 folds. Habit of playing with domestic animals increases the odds of protozoa infection by 3.82 folds. (Table 5)

**Table 4. The determinants of intestinal parasitic infection among adult household members (n = 2532).**

| Variable | | IP | | COR [95% CI] | AOR [95% CI] | p-value |
|---|---|---|---|---|---|---|
| | | Positive | Negative | | | |
| Sex | Male | 1079 | 266 | 0.07 [0.05–0.12] | 0.04 [0.02–0.09] | <0.01 |
| | Female | 1166 | 21 | | | |
| Environmental sanitation | Clean | 1280 | 107 | 2.23 [1.72–2.90] | 0.18 [0.12–0.27] | 0.01 |
| | Dirty | 965 | 180 | | | |
| Chicken | Present | 1454 | 63 | 6.54 [4.83–8.85] | 3.59 [2.38–5.41] | <0.01 |
| | Absent | 791 | 224 | | | |
| Role in the household | Children or mothers | 1277 | 39 | 8.39 [5.85–12.07] | 2.75 [1.51–4.99] | 0.01 |
| | Others | 968 | 248 | | | |
| Personal hygiene | Clean | 2113 | 270 | 1.01 [0.58–1.74] | 0.04 [0.01–0.12] | <0.01 |
| | Not clean | 132 | 17 | | | |
| Resident | Urban | 719 | 89 | 1.05 [0.8–1.38] | 2.32 [1.5–3.55] | <0.01 |
| | Rural | 1526 | 198 | | | |
| Substandard house | Yes | 946 | 108 | 1.21 [0.93–1.57] | 4.09[2.44–6.87] | <0.01 |
| | no | 1299 | 179 | | | |
| Source of light for the house | Traditional | 1692 | 247 | 0.5 [0.34–0.71] | 2.28 [1.19–4.37] | <0.01 |
| | Modern | 553 | 40 | | | |
| Family size | >4 | 1946 | 158 | 5.31 [4.05–6.97] | 7.18 [3.89–13.37] | <0.01 |
| | ≤4 | 299 | 129 | | | |
| Regular handwashing practice | Present | 208 | 2037 | 0.6 [0.41–0.87] | 0.4 [0.2–0.79] | <0.01 |
| | Absent | 42 | 245 | | | |

**Table 5. Specific predictors for soil transmitted helminths and protozoa infections.**

| Risk factors for | | | | | |
|---|---|---|---|---|---|
| Soil transmitted helminths | | | Protozoa infections | | |
| Variables | AOR [95% CI] | P-value | Variables | AOR [95% CI] | P-value |
| Barefoot | 1.51 [1.28–1.78] | <0.01 | Habit of ingesting raw vegetable | 2.96 [2.33–3.75] | <0.01 |
| Floor | 2.1 [1.81–2.44] | <0.01 | Habit of playing with domestic animals | 3.82 [3.17–4.61] | <0.01 |
| | | | Water source | 0.8 [0.68–0.95] | <0.01 |
| | | | Water filter | 0.65 [0.55–0.76] | <0.01 |

## Discussion

The prevalence of intestinal parasitic infection among family members of known intestinal parasitic case was 86.14% [95% CI: 85.12% - 87.15%]. The prevalence of intestinal parasitic infection among children family members was 82.77% [95% CI: 81.08% -84.47%]. The prevalence of intestinal parasitic infection among household members whose age greater than 16 years was 88.67% [95% CI: 87.43% -89.90%]. This result was in line with finding from Sudan and Central African Republic (95% CI for prevalence 78.69% -88.23%) [20, 21]. However, these results were higher than finding from Uganda (Prevalence of 55.04%) [22], and England (Prevalence of 30%) [23]. This might be due to the difference in living conditions. Our study area contains numerous contacts which increase the risk of acquiring intestinal parasitic infections.

The odds of intestinal parasitic infections among female household members were 24% higher during childhood and 96% higher during adulthood. This finding agrees with other scholar works [24]. This is due to the fact that women in the household are responsible to care for the child and disposal of the waste of the child which increases their risk of acquiring infection easily [25].

Environmental sanitation decreases the odds of intestinal parasitic infection by 96% during childhood, and by 82% during adulthood. This finding agrees with finding from other parts of Ethiopia [26]. This is because environmental sanitation eliminates the reservoir for intestinal parasitic infection which finally blocks the infectious cycle of the parasites [27].

The odds of intestinal parasitic infection were 2.75 higher in children and mothers as compared to the other household members. This finding agrees with findings from Accra [28]. This is because of the proximity of mothers and children to the household wastes, which contains numerous intestinal parasites [29].

The odds of intestinal parasitic infections were 2.68 folds higher among urban children, and 2.32 folds higher in urban adults. This finding agrees with findings from India [30]. This might be due to poor environmental sanitation conditions with the overcrowding situation in the urban area [31].

Personal hygiene decreases the odds of intestinal parasitic infection by 74% among children, and by 96% lower in adults. This finding agrees with systematic review report [32]. This is because personal hygiene breaks the chain of intestinal parasitic transmission cycle [33].

Living in the substandard housing condition increases the odds of intestinal parasitic infection by 1.92 folds higher in children, and by 4 folds higher in adults. This finding agrees with finding from Brazil [34]. This is because of better sanitary facility access of the group [35].

The odds of intestinal parasitic infection were 2.28 folds higher among household members using traditional light for their house. This finding agrees with clinical trial results [36]. This is because if the household was supplied with electricity, the household members can become aware of a health- related condition thought radio, television mass education which finally increases their awareness of a health- related condition.

Regular handwashing practice decreases the odds of intestinal parasitic infection by 60%. This finding was in line with 2018 finding from Ethiopia [37]. This is because the feco-oral route of transmission will be blocked by applying regular handwashing practice [38].

Higher family size increases the odds of intestinal parasitic infection by 7.18 folds. This finding agrees with the previous finding from the same study area [39]. This is because high family size decreases the access to the basic sanitary facility due to sharing of the limited resources.

The presence of chicken in the house increases the odds of intestinal parasitic infection by 4.42 folds higher in children, and by 3.39 folds higher in adults. This finding agrees with findings from China [40]. This is because chickens act as a reservoir to numerous intestinal parasite species [41].

The presence of household water filtering materials decreases the odds of protozoa infection by 35%. This finding agrees with systematic review pools across the globe [42]. This is because of water treatment at the households levels eliminates the eggs or cysts of protozoa from the water [43].

Habit of playing with a domestic animal increases the odds of protozoa infection by 3.82 folds. This finding agrees with finding from Canada [44]. This is because most protozoa infections are transmitted from animals to humans (zoonotic) [45].

Using pipe water decreases the odds of protozoa infection by 20%. This finding agrees with finding from Brazil [46]. This indicated that untreated water is a potential source of protozoa infection [47].

Barefoot behavior increases the odds of soil-transmitted helminths infection by 4.5 folds. This finding was in line with 2018 results from Nigeria [48]. This is because barefoot allows the entry of soil transmitted helminths like hookworm at its infective stage [49].

The odds of soil-transmitted helminths were 2 folds higher in individuals living in a house made from the mud floor. This finding agrees with finding from Kenya [50]. This is because most people prefer barefoot in the house which increases the risk of soil-transmitted helminths.

The main limitation of this study was a failure to identify the incident and prevalent cases, but the overall aim of this study was to estimate the prevalence of intestinal parasitic infection among household members mixing of new or pre-existing cases will not create a huge problem.

## Conclusion

The prevalence of intestinal parasites was high among household contacts of intestinal parasite-infected family members. Intestinal parasitic infection among household members was determined by family size, environmental sanitation, substandard housing, gender, household water treatment, habit of playing with domestic animals, The presence of chicken in the house, source of water, role in the household, resident, source of light, handwashing practice, and barefoot.

## Recommendation

Clinicians must trace and care for all household contacts of intestinal parasite patients to make the interventions effective at the community level.

## Supporting information

**S1 Questionnaire. The data collection tool (questionnaire for the study).**
(DOCX)

**S1 Data. The SPSS data file of the research.**
(SAV)

## Acknowledgments

Our heartfelt acknowledgment goes to household members for good cooperation during the fieldwork. We would also like to acknowledge Mecha district health office for their unreserved efforts. At last but not least we would also like to acknowledge all organization and individuals that contributed to this research work.

## Author Contributions

**Conceptualization:** Berhanu Elfu Feleke, Melkamu Bedimo Beyene, Teferi Elfu Feleke, Tadesse Hailu Jember, Bayeh Abera.

**Data curation:** Berhanu Elfu Feleke, Melkamu Bedimo Beyene, Teferi Elfu Feleke, Tadesse Hailu Jember, Bayeh Abera.

**Formal analysis:** Berhanu Elfu Feleke, Melkamu Bedimo Beyene, Teferi Elfu Feleke, Bayeh Abera.

**Funding acquisition:** Berhanu Elfu Feleke, Melkamu Bedimo Beyene, Tadesse Hailu Jember, Bayeh Abera.

**Investigation:** Berhanu Elfu Feleke, Melkamu Bedimo Beyene, Teferi Elfu Feleke, Tadesse Hailu Jember, Bayeh Abera.

**Methodology:** Berhanu Elfu Feleke, Melkamu Bedimo Beyene, Teferi Elfu Feleke, Tadesse Hailu Jember, Bayeh Abera.

**Project administration:** Berhanu Elfu Feleke, Melkamu Bedimo Beyene, Tadesse Hailu Jember.

**Resources:** Berhanu Elfu Feleke, Melkamu Bedimo Beyene, Tadesse Hailu Jember, Bayeh Abera.

**Software:** Berhanu Elfu Feleke, Melkamu Bedimo Beyene.

**Supervision:** Berhanu Elfu Feleke, Melkamu Bedimo Beyene, Teferi Elfu Feleke, Tadesse Hailu Jember.

**Validation:** Berhanu Elfu Feleke, Melkamu Bedimo Beyene, Teferi Elfu Feleke, Tadesse Hailu Jember.

**Visualization:** Berhanu Elfu Feleke, Teferi Elfu Feleke.

**Writing – original draft:** Berhanu Elfu Feleke, Melkamu Bedimo Beyene, Teferi Elfu Feleke, Tadesse Hailu Jember, Bayeh Abera.

**Writing – review & editing:** Berhanu Elfu Feleke, Melkamu Bedimo Beyene, Teferi Elfu Feleke, Tadesse Hailu Jember, Bayeh Abera.

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
