## [Decision Letter · Decision Letter 0]

29 Aug 2019

[EXSCINDED]

PONE-D-19-20993

Intestinal parasitic infection among household contacts of primary cases, a comparative cross-sectional study

PLOS ONE

Dear Mr, Feleke,

Thank you for submitting your manuscript to PLOS ONE. After careful consideration, we feel that it has merit but does not fully meet PLOS ONE’s publication criteria as it currently stands. Therefore, we invite you to submit a revised version of the manuscript that addresses the points raised during the review process.

ACADEMIC EDITOR:In this manuscript, the authors tried to estimate the prevalence and determinants of intestinal parasitic infection among family members of known intestinal parasitic infected patients in Mecha district of Ethiopia. The manuscript has merit because the information is helpful to develop possible strategy for parasite prevention in Ethiopia. Although the study design is straightforward, but the content somewhat is too simple and not in-depth enough. Therefore, this article cannot be accepted for publication in current form d except some serious concerns have been clarified.

We would appreciate receiving your revised manuscript by Oct 13 2019 11:59PM. To enhance the reproducibility of your results, we recommend that if applicable you deposit your laboratory protocols in protocols.io, where a protocol can be assigned its own identifier (DOI) such that it can be cited independently in the future. For instructions see: http://journals.plos.org/plosone/s/submission-guidelines#loc-laboratory-protocols

We look forward to receiving your revised manuscript.

Kind regards,

Chia Kwung Fan, LL.M, PhD

Academic Editor

PLOS ONE

Journal Requirements:

2. Please include in your financial disclosure statement the name of the funders of this study (as well as grant numbers if available).

3. Please specify whether an interview guide was used to interview the participants in your study. If yes, please describe and/or include a copy as a Supporting Information file.

Reviewers' comments:

Reviewer's Responses to Questions

**Comments to the Author**

1. Is the manuscript technically sound, and do the data support the conclusions?

Reviewer #1: No

Reviewer #2: Partly

2. Has the statistical analysis been performed appropriately and rigorously? 

Reviewer #1: Yes

Reviewer #2: Yes

3. Have the authors made all data underlying the findings in their manuscript fully available?

Reviewer #1: Yes

Reviewer #2: Yes

4. Is the manuscript presented in an intelligible fashion and written in standard English?

Reviewer #1: Yes

Reviewer #2: Yes

5. Review Comments to the Author

Reviewer #1: The study of intestinal parasitic infection among household contacts of primary cases, a comparative cross-sectional study is interesting; however, some major concerns should be clarified before its suitable publication in PLOS ONE.

1.Result: How do you make a confirmative diagnosis of Enatmeba histolytic (11.4%)? By what kind of criteria?

2.Discussion: line 167-Would you please compare the similarity or difference between your study and the other African countries, instead England only?

3.Line 174: Authors indicated that “environmental sanitation decreases the odds of intestinal parasitic infection by 96% during childhood, and by 82% during adulthood”, how can you get such an estimation? The other similar calculation should be indicated also.

Reviewer #2: In this manuscript, the authors tried to estimate the prevalence and determinants of intestinal parasitic infection among family members of known intestinal parasitic infected patients in Mecha district of Ethiopia. The manuscript has merit because the information is helpful to develop possible strategy for parasite prevention in Ethiopia. Although the study design is straightforward, but the content somewhat is too simple and not in-depth enough. Therefore, this article cannot be published except some serious problems have been addressed. These issues are discussed below.

1. Major problem is that the authors put all the intestinal parasite infection in together for survey without discussing the differences between all these distinct species, especially their infection routes. For example, hookworm and S. mansoni are infected by invading through skin; so the considered determinant parameters that should be different with the parasites of fecal-oral infection. However, authors here combine all these parasitic infection as a topic and compare to the same determinants. It would be also wrong to put protozoa and helminth as a simple field to discuss, because the inspection level and infection way are very different, such as soil- or water-transmission, reservoir host or not; it should be given more detailed division for considering their determinants. The influence of each parameter on different parasites cannot be the same.

2. A lot of the parameters of tables are unclear and not accurate enough, e.g. "Environmental sanitation" only presents as clean or dirty, it should be shown with different levels for distinguishing. Additionally, the lack of in-death enough discussion about their data make the author’s descriptions in results or discussion section are too general and fuzzy, like a hodgepodge. It cannot show the relevance of their results as the hypothesis of this manuscript, even worse to arrive at the stated conclusions. The significance of these determinants for intestinal parasitic infection is not hand out.

3. In table 2, what kinds of parasite are included in the mixed infection? The patient is infected with two or three parasites? These data may be important to ascertain the source of dissemination and influenced determinants.

4. Authors should attach the Ethical approval number or the document by ethical committee for reference.

5. The authors analyzed the determinants of intestinal parasite infections in adult and child household members, but do not make any further explain between their difference and the reasons? Some determinants described in discussion section are unreasonable, e.g. the source of light is not relevant to the awareness of a health- related condition.

6. The author's recommendation is not specific enough and lacks order. Which determinant is the most important?

6. PLOS authors have the option to publish the peer review history of their article (what does this mean?). If published, this will include your full peer review and any attached files.

Reviewer #1: No

Reviewer #2: No

---

## [Author Response · Author response to Decision Letter 0]

5 Sep 2019

Response to the reviewers

Manuscript title: Intestinal parasitic infection among household contacts of primary cases, a comparative cross-sectional study

Manuscript ID: PONE-D-19-20993

Reviewer #1: The study of intestinal parasitic infection among household contacts of primary cases, a comparative cross-sectional study is interesting; however, some major concerns should be clarified before its suitable publication in PLOS ONE.

1. Result: How do you make a confirmative diagnosis of Enatmeba histolytic (11.4%)? By what kind of criteria? 

Response: Sorry it was the type error, Enatmeba histolytic was revised with Enatmeba histolytica/ dispar since we used microscopic stool examination. Result section, line number 122 -126 and table 2, page 8

• 2.Discussion: line 167-Would you please compare the similarity or difference between your study and the other African countries, instead England only?

• Response: revised, the finding was compared from African countries (Sudan, central African republic and Uganda ), discussion section page 16, line number 167-168

3.Line 174: Authors indicated that “environmental sanitation decreases the odds of intestinal parasitic infection by 96% during childhood, and by 82% during adulthood”, how can you get such an estimation? The other similar calculation should be indicated also.

• Response: when the odds ratio is less than 1, the protective fraction will be calculated by subtracting from 1. Result section page 10, table 3 shows that the association between environmental sanitation and IP infection was protective, i.e. AOR is less than 1(0.04), the preventive fraction was calculated by subtracting from (1 - 0.04=0.96, meaning environmental sanitation is 96 % protective). Table 4, page 13 presents the association between environmental sanitation and IP infection in adults was 0.18, the preventive fraction (1-0.18) = 82% protective. 

Reviewer #2: In this manuscript, the authors tried to estimate the prevalence and determinants of intestinal parasitic infection among family members of known intestinal parasitic infected patients in Mecha district of Ethiopia. The manuscript has merit because the information is helpful to develop possible strategy for parasite prevention in Ethiopia. Although the study design is straightforward, but the content somewhat is too simple and not in-depth enough. Therefore, this article cannot be published except some serious problems have been addressed. These issues are discussed below.

1. Major problem is that the authors put all the intestinal parasite infection in together for survey without discussing the differences between all these distinct species, especially their infection routes. For example, hookworm and S. mansoni are infected by invading through skin; so the considered determinant parameters that should be different with the parasites of fecal-oral infection. However, authors here combine all these parasitic infection as a topic and compare to the same determinants. It would be also wrong to put protozoa and helminth as a simple field to discuss, because the inspection level and infection way are very different, such as soil- or water-transmission, reservoir host or not; it should be given more detailed division for considering their determinants. The influence of each parameter on different parasites cannot be the same.

• Response: the common predictors for all intestinal parasites like sex, environmental sanitation, role in the household, residence, personal hygiene, age, living in substandard house, regular handwashing practice and family size was presented together. These predictors are common for soil-transmitted helminths and protozoa infections. The specific predictors for protozoa(habit of ingesting raw vegetable, playing with domestic animals, water source, the presence of water filtering materials) and soil transmitted helminths(barefoot, floor materials) were presented in result section page 15, table 5 and well discussed in discussion section page 18, line number 208-223. 

• 2. A lot of the parameters of tables are unclear and not accurate enough, e.g. "Environmental sanitation" only presents as clean or dirty, it should be shown with different levels for distinguishing. Additionally, the lack of in-death enough discussion about their data make the author’s descriptions in results or discussion section are too general and fuzzy, like a hodgepodge. It cannot show the relevance of their results as the hypothesis of this manuscript, even worse to arrive at the stated conclusions. The significance of these determinants for intestinal parasitic infection is not hand out.

• Response: the specific predictors for soil-transmitted helminths and protozoa were presented in table 5(result section, page 15) and discussed in discussion section page 18. But there are common risk factors for intestinal parasites like environmental sanitation, residence, sex, age, living in substandard housing conditions, role in the household and family size. These factors are common for all protozoa and soil transmitted helminths. 

• 3. In table 2, what kinds of parasite are included in the mixed infection? The patient is infected with two or three parasites? These data may be important to ascertain the source of dissemination and influenced determinants.

• Response: mixed infection includes study participants infected by two or more parasites. 

4. Authors should attach the Ethical approval number or the document by ethical committee for reference.

• Response: Attached, the ethical approval number for this research was “የአ.ህ.ጤቴ/ሽ/ዳ/01/795” methods section page 7, line number 109. 

5. The authors analyzed the determinants of intestinal parasite infections in adult and child household members, but do not make any further explain between their difference and the reasons? Some determinants described in discussion section are unreasonable, e.g. the source of light is not relevant to the awareness of a health- related condition.

• Response: it is commonly observed that if the house contains electricity, the probability of having Television and radio is very high which increases their awareness towards health related conditions because of mass health education given from the broadcast. 

6. The author's recommendation is not specific enough and lacks order. Which determinant is the most important?

• Response: the prevalence of intestinal parasitic infection among children household members was 82% and the prevalence of intestinal parasite infection among adult household members was 86%. The intestinal parasitic infection intervention currently did not touch the contact tracing, the main message of this work is that clinicians was missing more than 80% of intestinal parasitic infection cases. So they should incorporate contact tracing.

---

## [Decision Letter · Decision Letter 1]

19 Sep 2019

PONE-D-19-20993R1

Intestinal parasitic infection among household contacts of primary cases, a comparative cross-sectional study

PLOS ONE

Dear Mr, Feleke,

Thank you for submitting your manuscript to PLOS ONE. After careful consideration, we feel that it has merit but does not fully meet PLOS ONE’s publication criteria as it currently stands. Therefore, we invite you to submit a revised version of the manuscript that addresses the points raised during the review process.

ACADEMIC EDITOR: Although the authors have answered the reviewer's questions appropriately, however, minor errors should be amended before its suitable publication.

1.Table 2: intestinal parasites—the parasite name should be e.g., Ascaris lumbricoides instead Ascaris Lumbricoides. The first letter of the species should be lowercase. This indication

2.line 167: This results was in line with finding from Sudan 168 and central African republic [20, 21], higher than finding from Uganda [22], and England [23].

Please show their prevalence, respectively. Also, please check the grammar and spellings.

We would appreciate receiving your revised manuscript by Nov 03 2019 11:59PM. To enhance the reproducibility of your results, we recommend that if applicable you deposit your laboratory protocols in protocols.io, where a protocol can be assigned its own identifier (DOI) such that it can be cited independently in the future. For instructions see: http://journals.plos.org/plosone/s/submission-guidelines#loc-laboratory-protocols

We look forward to receiving your revised manuscript.

Kind regards,

Chia Kwung Fan, LL.M, PhD

Academic Editor

PLOS ONE

Additional Editor Comments (if provided):

Although the authors have answered the reviewer's questions appropriately, however, minor errors should be amended before its suitable publication.

1.Table 2: intestinal parasites—the parasite name should be e.g., Ascaris lumbricoides instead Ascaris Lumbricoides. The first letter of the species should be lowercase. This indication

2.line 167: This results was in line with finding from Sudan 168 and central African republic [20, 21], higher than finding from Uganda [22], and England [23].

Please show their prevalence, respectively. Also, please check the grammar and spellings.

Reviewers' comments:

Reviewer's Responses to Questions

**Comments to the Author**

1. If the authors have adequately addressed your comments raised in a previous round of review and you feel that this manuscript is now acceptable for publication, you may indicate that here to bypass the “Comments to the Author” section, enter your conflict of interest statement in the “Confidential to Editor” section, and submit your "Accept" recommendation.

Reviewer #1: (No Response)

Reviewer #2: All comments have been addressed

2. Is the manuscript technically sound, and do the data support the conclusions?

Reviewer #1: Yes

Reviewer #2: Yes

3. Has the statistical analysis been performed appropriately and rigorously? 

Reviewer #1: Yes

Reviewer #2: Yes

4. Have the authors made all data underlying the findings in their manuscript fully available?

Reviewer #1: Yes

Reviewer #2: Yes

5. Is the manuscript presented in an intelligible fashion and written in standard English?

Reviewer #1: Yes

Reviewer #2: Yes

6. Review Comments to the Author

Reviewer #1: Although the authors have answered the reviewer's questions appropriately, however, minor errors should be amended before its suitable publication.

1.Table 2: intestinal parasites—the parasite name should be e.g., Ascaris lumbricoides instead Ascaris Lumbricoides. The first letter of the species should be lowercase. This indication

2.line 167: This results was in line with finding from Sudan 168 and central African republic [20, 21], higher than finding from Uganda [22], and England [23].

Please show their prevalence, respectively. Also, please check the grammar and spellings.

Reviewer #2: In this manuscript, the authors tried to estimate the prevalence and determinants of intestinal parasitic infection among family members of known intestinal parasitic infected patients in Mecha district of Ethiopia. The manuscript has merit because the information is helpful to develop possible strategy for parasite prevention in Ethiopia. All the comments have been addressed.

7. PLOS authors have the option to publish the peer review history of their article (what does this mean?). If published, this will include your full peer review and any attached files.

Reviewer #1: No

Reviewer #2: No

---

## [Author Response · Author response to Decision Letter 1]

19 Sep 2019

Response to the reviewers

Although the authors have answered the reviewer's questions appropriately, however, minor errors should be amended before its suitable publication.

1.Table 2: intestinal parasites—the parasite name should be e.g., Ascaris lumbricoides instead Ascaris Lumbricoides. The first letter of the species should be lowercase. 

• Response: revised based on the comment 

This results was in line with finding from Sudan and central African republic [20, 21], higher than finding from Uganda [22], and England [23].

Please show their prevalence, respectively. Also, please check the grammar and spellings.

• Response: revised based on the comments.

---

## [Decision Letter · Decision Letter 2]

25 Sep 2019

Intestinal parasitic infection among household contacts of primary cases, a comparative cross-sectional study

PONE-D-19-20993R2

Dear Dr. Feleke,

We are pleased to inform you that your manuscript has been judged scientifically suitable for publication and will be formally accepted for publication once it complies with all outstanding technical requirements.

With kind regards,

Chia Kwung Fan, LL.M, PhD

Academic Editor

PLOS ONE

Additional Editor Comments (optional):

Reviewers' comments:

Reviewer's Responses to Questions

**Comments to the Author**

1. If the authors have adequately addressed your comments raised in a previous round of review and you feel that this manuscript is now acceptable for publication, you may indicate that here to bypass the “Comments to the Author” section, enter your conflict of interest statement in the “Confidential to Editor” section, and submit your "Accept" recommendation.

Reviewer #1: All comments have been addressed

2. Is the manuscript technically sound, and do the data support the conclusions?

Reviewer #1: Yes

3. Has the statistical analysis been performed appropriately and rigorously? 

Reviewer #1: Yes

4. Have the authors made all data underlying the findings in their manuscript fully available?

Reviewer #1: Yes

5. Is the manuscript presented in an intelligible fashion and written in standard English?

Reviewer #1: Yes

6. Review Comments to the Author

Reviewer #1: The authors have answered all the queries appropriately thus the reviewer recommends it to be accepted for publication in PLOS ONE.

7. PLOS authors have the option to publish the peer review history of their article (what does this mean?). If published, this will include your full peer review and any attached files.

Reviewer #1: No

---

## [Editor Report · Acceptance letter]

30 Sep 2019

PONE-D-19-20993R2 

Intestinal parasitic infection among household contacts of primary cases, a comparative cross-sectional study 

Dear Dr. Feleke:

I am pleased to inform you that your manuscript has been deemed suitable for publication in PLOS ONE. Congratulations! Your manuscript is now with our production department. 

With kind regards,

on behalf of

Dr. Chia Kwung Fan 

Academic Editor

PLOS ONE